# Graft Failure Due to Nonadherence among 150 Prospectively-Followed Kidney Transplant Recipients at 18 Years Post-transplant: Our Results and Review of the Literature

**DOI:** 10.3390/jcm11051334

**Published:** 2022-02-28

**Authors:** Jeffrey J. Gaynor, Giselle Guerra, David Roth, Linda Chen, Warren Kupin, Adela Mattiazzi, Mariella Ortigosa-Goggins, Marina M. Tabbara, Lissett Moni, George W. Burke, Gaetano Ciancio

**Affiliations:** 1Department of Surgery, Miami Transplant Institute, University of Miami Miller School of Medicine, Miami, FL 33136, USA; ljchen@med.miami.edu (L.C.); mmt68@med.miami.edu (M.M.T.); ltueros@med.miami.edu (L.M.); gburke@med.miami.edu (G.W.B.III); gciancio@med.miami.edu (G.C.); 2Department of Medicine, Miami Transplant Institute, University of Miami Miller School of Medicine, Miami, FL 33136, USA; gguerra@med.miami.edu (G.G.); d.roth@med.miami.edu (D.R.); wkupin@med.miami.edu (W.K.); amattiazzi@med.miami.edu (A.M.);

**Keywords:** kidney transplantation, 18-year prospective cohort study, cause-specific graft loss estimates, graft failure due to nonadherence, overt nonadherence, prognostic factors

## Abstract

Background: We previously reported that graft failure due to nonadherence (GFNA) was a major cause of graft loss in kidney transplantation. Here, among 150 prospectively-followed kidney transplant recipients at 18 years post-transplant, we provide: updated (longer-term) estimates of cause-specific graft loss probabilities, risk factors for developing GFNA, and detailed characterizations of patients’ overt nonadherent (NA) behavior, including timing, extent, and clinical consequences. Methods: Determination of the patient becoming NA in taking his/her immunosuppressive medications, and the underlying cause of graft loss, were determined prospectively by the attending physicians. For never-functioning-graft, GFNA, GF due to causes other than NA (Other GF), and death with a functioning graft (DWFG), cumulative incidence functions were used to estimate the cumulative probabilities of cause-specific graft loss. Cox stepwise regression was used to determine significant multivariable predictors for the hazard rate of developing GFNA. Results: GFNA was a major cause of graft loss (22/150 patients), particularly among African-American and Hispanic recipients <50 years of age-at-transplant (20/56 experienced GFNA), with estimated percentages of such patients ever developing GFNA ranging between 36.9 and 41.5%. These patients were also at a higher risk of developing Other GF. For the remaining patients (2/94 experienced GFNA), estimated percentages of ever-developing GFNA were much lower (range: 0.0–6.7%). The major cause of graft loss among recipients ≥50 years of age was DWFG; GFNA rarely occurred among older recipients. In 21/22 GFNA patients, NA behavior lasted continuously from the time of developing NA until GFNA. In total, 28/150 patients became NA, and 67.9% (19/28) occurred beyond 36 months post-transplant. A total of 25 of 28 NA patients (89.3%) developed biopsy-proven acute rejection and/or chronic rejection that was directly attributed to the NA behavior. Lastly, 25/28 admitted to NA behavior, with financial and psychological components documented in 71.4% (20/28) and 96.4% (27/28) of NA cases, respectively. Conclusions: These results highlight the importance of performing serial monitoring of patients for overt NA behavior throughout their post-transplant follow-up. Financial and psychological components to NA behavior need to be simultaneously addressed with the goal of achieving complete avoidance/elimination of NA behavior among higher risk patients.

## 1. Introduction

In 2014, we reported that graft failure due to nonadherence (GFNA) (nonadherence being defined as the patient either abruptly stopping or consistently not taking the prescribed immunosuppressive medications) was a major cause of graft loss among 628 adult primary kidney transplant recipients, having a median follow-up of 56 months post-transplant [1], results that were consistent with a number of other (mostly older) reports [2,3,4,5,6,7,8,9,10]. In that study, we pooled kidney transplant recipients across four randomized immunosuppression trials performed at our center since 2000, with the goal of maximizing statistical power and precision. We recently reported the results at 18 years post-transplants for the 150 study participants in the first trial [11], which included 10 additional years of patient follow-up compared with our previous reports [1,12]. It is relatively rare for longer-term estimates of cause-specific graft loss following kidney transplantation to be reported, particularly clinical trial results occurring beyond 8–10 years post-transplant.

Many reports of patient nonadherence (NA) in the literature have focused on analyzing the results of patient self-reporting using the strict dichotomy, “whether any immunosuppression-taking NA has occurred”, with “no” implying perfect adherence, and “yes” implying any evidence of nonadherent (NA) behavior [13,14,15,16,17,18,19,20,21,22,23,24,25,26,27,28,29,30]. The main problem with using such a strict dichotomy of “perfect adherence” vs. “any type of NA behavior” is the fact that this NA definition includes all levels of minor/subclinical NA behavior, including patients who (to a small degree) mistime their dose-taking, take less than the prescribed dose, and/or even skip taking some doses. A more revealing type of NA behavior, we believe, which can be captured by patient self-reporting, and was defined by Rosenberger et al. [15] as a “major noncompliance” and by Dunn et al. [9] as “overt NA”, focuses on whether or not the patient has abruptly stopped taking, or has consistently not been taking, the immunosuppressive medications as prescribed, in such a way that leads directly to clinical consequences, i.e., rejection and/or graft failure. Specifically, Rosenberger et al. [15] defined “major noncompliance” as “the situation when a patient dramatically violates the immunosuppressive regime with following rejection episode and graft loss as a consequence,” and Dunn et al. [9] defined “overt nonadherence” as the patient “admitting to not taking the immunosuppressive medications for a prolonged period of time.” These more overt definitions of NA behavior are similar to those used in previous reports and by our center [1,2,3,4,5,6,7,8,10]. In terms of those patients who became overtly NA, Dunn et al. [9] added, “Many of these recipients also did not follow laboratory test protocols, did not keep clinic appointments, or did not communicate with our transplant center at some point after their transplant,” which is, again, in agreement with what we previously observed [1].

Therefore, using this prospectively followed cohort of 150 patients with extended (18 years of post-transplant) follow-up, and in keeping with our previous approach to identify patients who became “overtly” NA, [1] we provide here an updated, longer-term analysis of: (i) the estimated probabilities of cause-specific graft loss, for all patients combined, as well as stratified by younger vs. older recipient age (at transplant), and (ii) risk factors for developing GFNA, knowing from our previous report [1] that African-American and Hispanic recipients <50 years of age were at a particularly higher risk of experiencing GFNA. We also attempted to provide a more accurate characterization of those patients who became overtly NA, including estimated times (from transplant) to becoming NA, the total length of time that the patient demonstrated NA behavior, and whether or not (i) the NA behavior led directly to rejection and/or GFNA, (ii) the patient admitted to his/her NA behavior, (iii) the patient had documented immunosuppressive trough levels that were either low or undetectable, and (iv) there were indicated/acknowledged reason(s) for the patient’s NA behavior.

## 2. Materials and Methods

### 2.1. Data Description

As previously reported, [11,12,31,32,33] between May 2000 and December 2001 a prospective randomized trial of 150 (living and deceased donor) adult primary kidney transplant recipients was performed, comparing tacrolimus/sirolimus vs. tacrolimus/mycophenolate mofetil vs. cyclosporine microemulsion/sirolimus (50 per arm). The center institutional review board approved the protocol; all patients gave written informed consent prior to enrollment (Clinical Trials.gov ID: NCT00681213). In addition, the clinical and research activities being reported adhere to the Declaration of Helsinki and are consistent with the Principles of the Declaration of Istanbul as outlined in the ‘Declaration of Istanbul on Organ Trafficking and Transplant Tourism’. Details of the planned immunosuppression, as well as mean immunosuppressive drug dosing and trough levels, achieved in each arm over time are omitted here, as this information and the major findings of this randomized trial were recently reported [11]. Details regarding current immunosuppression, drug dose changes, protocol violations, and reasons for those changes/protocol violations were also summarized in our most recent report and are omitted here [11]. The last follow-up date in our most recent report [11] was 1 June 2019; thus, median follow-up among patients who were still alive with a functioning graft was 221 months (range: 210–228 months) (18.4 years) post-transplant.

Nonadherence (NA) was defined as the recipient abruptly stopping or consistently not taking the prescribed immunosuppressive medications, documented by (i) clinical notes of the attending physician(s) and nurse coordinator(s), (ii) patient (or guardian) admission, and (iii) maintenance drug trough levels [1,11,12]. Such determinations were based upon a detailed clinical history (taken by both the attending physician and nurse coordinator) of the patient failing to take his/her immunosuppressive medications as prescribed. In most cases the patient acknowledged becoming NA. NA was also corroborated by serially obtained immunosuppressive drug levels at or close to zero, confirming the attending physician’s determination of NA. The date of first becoming NA was approximated based on available information.

All graft loss causes, never functioning graft (NFG), GFNA, GF due to underlying causes other than NA (Other GF), and death with a functioning graft (DWFG), were determined prospectively from the attending physician’s ongoing clinical evaluation of each patient’s post-transplant follow-up. In each case where GFNA was determined as the underlying cause of graft failure, the attending physician concluded that the patient had become sufficiently NA in taking his/her immunosuppressive medications to trigger graft failure (return to permanent dialysis) [1].

### 2.2. Statistical Analysis

For the four distinct graft loss causes (i.e., NFG, GFNA, Other GF, and DWFG), Nelson–Aalen cause-specific cumulative hazard plots (rather than Kaplan–Meier curves) were used for graphical display of the cause-specific hazards, as the cause-specific hazard rate is more directly visualized by the slopes of the cumulative hazard curves. Conventional log-rank tests of subgroup differences for a particular cause-specific hazard were performed, whereby competing causes of failure were treated as censored observations.

The cumulative incidence function (CIF) [34,35] was used to estimate the cumulative probability of developing graft loss from one cause (by post-transplant time, t) in the presence of the other causes. Patient subgroup differences for a particular CIF were tested using the log-rank test, whereby competing causes of failure were treated as censored observations beyond the last observed failure time from the cause of interest (i.e., included in all risk sets) [36,37]. Given that NA occurrence was of particular interest in this study, Cox stepwise regression was also used to determine a multivariable set of significant predictors for the hazard rate of (i) becoming NA, (ii) developing GFNA, and (iii) the CIF for GFNA (i.e., the sub-distribution hazard rate of developing GFNA). *P*
< 0.05 were considered as statistically significant.

Finally, while we knew from our previous report [1] that African-American and Hispanic recipients <50 years of age were at particularly higher risk of experiencing GFNA, recipient age was treated as a continuous variable in the Cox models. Stratification of recipient age by <50 vs. ≥50 years was used primarily for graphical display. In addition, as previously shown, the risk of experiencing GFNA was similarly higher among African-American and Hispanic recipients, [1] and no significant differences in other clinical outcomes were observed between these two subgroups. Thus, African-American and Hispanic recipients were combined for the analysis shown here.

## 3. Results

### 3.1. Distributions of Selected Baseline Variables

Distributions of selected baseline variables appear in Table 1. Median recipient age (at transplant) was 48 years (range: 14–78 years); 66.7% (100/150) were male. Black (non-Hispanic), Hispanic, and White (non-Hispanic) race/ethnicity comprised 20.7% (31/150), 37.3% (56/150), and 42.0% (63/150) of the recipients, respectively.

### 3.2. Cause-Specific Graft Loss for All Patients Combined and Stratified by Recipient Age

A cause-specific cumulative hazard plot for all patients combined is presented in Figure 1A. Other than NFG, which only occurred in four patients during the first 2 weeks post-transplant, the hazard rates of GFNA, Other GF, and DWFG (i.e., slopes of the cumulative hazard curves) were reasonably constant over time beyond 5 years post-transplant (with 22, 31, and 40 patients experiencing these cause-specific graft losses, respectively). Thus, there did not appear to be any falloff in these cause-specific hazard rates over time, even at 18 years post-transplant.

Cause-specific cumulative hazard plots for patients stratified by recipient age (at transplant) <50 (N = 82) vs. ≥50 (N = 68) years are presented in Figure 1B,C, respectively. Here, one can see that the cause-specific patterns of failure were dramatically different between the younger and older patients. First, GFNA and Other GF were the primary causes of graft loss for patients <50 years of age at transplant, particularly during the first 10 years post-transplant, whereas DWFG was the primary cause of graft loss for patients ≥50 years of age at transplant. The observed proportions of graft loss due to GFNA, Other GF, and DWFG for patients <50 years of age at transplant were 21/82, 21/82, and 8/82, respectively, in comparison with 1/68, 10/68, and 32/68 for patients ≥50 years of age at transplant, respectively. Thus, the hazard rates of developing GFNA and Other GF were distinctly higher for patients <50 years of age at transplant, and the log-rank test comparing death-censored graft survival (combining NFG, GFNA, and Other GF as one outcome) between recipient age <50 and ≥50 years yielded *p* = 0.002. Conversely, the hazard rate of DWFG was distinctly higher for patients ≥50 years of age, with the log-rank test comparing the hazard rate of DWFG between recipient age <50 and ≥50 years yielding *p* < 0.000001. Second, the hazard rate of developing GFNA was clearly higher for patients <50 years of age at transplant in comparison with patients ≥50 years of age (*p* = 0.0005), with GFNA occurring in only one patient ≥50 years of age. Third, Figure 1B shows that in patients <50 years of age, the hazard rates of GFNA and Other GF were of similar magnitude (and both higher when compared with their respective hazard rates in Figure 1C); thus, even if GFNA were to become eliminated as a cause of graft loss, the younger patients (compared with older patients) would remain at higher risk of developing Other GF over time.

CIF estimates (±SE) of the percentages experiencing cause-specific graft loss for all patients combined and then stratified by recipient age <50 vs. ≥50 years appear in Table 2 and Figure 2A–C, respectively. For all patients combined, estimates at 216 months (18 years) post-transplant for NFG, GFNA, Other GF, and DWFG were 2.7% ± 1.3%, 16.9% ± 3.4%, 23.8% ± 3.8%, and 30.2% ± 4.1%, respectively. Of note, among the 22 patients who experienced GFNA, only 13.6% (3/22) occurred during the first 36 months post-transplant. For patients <50 years of age at transplant, estimates at 216 months (18 years) post-transplant for NFG, GFNA, Other GF, and DWFG were 3.7% ± 2.1%, 28.8% ± 5.4%, 29.3% ± 5.4%, and 11.3% ± 3.8%, respectively. In comparison, for patients ≥50 years of age at transplant, estimates at 216 months (18 years) post-transplant for NFG, GFNA, Other GF, and DWFG were 1.5% ± 1.5%, 1.7% ± 1.7%, 15.1% ± 4.4%, and 55.1% ± 7.2%, respectively. The log-rank test for a difference in the CIF for GFNA between recipient age <50 and ≥50 years was highly significant, yielding *p* = 0.00002. While the log-rank test for a difference in the CIF for Other GF between recipient age <50 vs. ≥50 years was not statistically significant, yielding *p* = 0.13, the log-rank test for a difference in the CIF for DWFG between recipient age <50 and ≥50 years was highly significant, yielding *p* < 0.000001.

### 3.3. Multivariable Analyses of GFNA

Results of the Cox stepwise regression analyses for the hazard rates of developing GFNA and the CIF for GFNA (i.e., subdistribution hazard) were essentially the same in that two factors were jointly associated with a significantly higher GFNA rate: younger recipient age (as a continuous variable) (*p* = 0.0001 for the GFNA hazard rate; *p* = 0.00003 for the GFNA subdistribution hazard) and African-American or Hispanic recipient (*p* = 0.0007 for the GFNA hazard rate; *p* = 0.001 for the GFNA subdistribution hazard) (Cox model coefficients and corresponding SEs are shown in Table 3). For both Cox models, no other baseline variable was significant in either univariable or multivariable analysis, including the treatment arm of maintenance immunosuppression in which the patient had been assigned (results not shown).

The multivariable influence of recipient age (categorized by <35, 35–49, and ≥50 years) and race/ethnicity (African-American or Hispanic vs. non-African-American and non-Hispanic) are reflected by the cumulative hazard plots and CIF estimates shown in Figure 3A,B, respectively. CIF estimates (±SE) of the percentages experiencing GFNA stratified by these two baseline variables are also shown in Table 4. The two higher risk groups were clearly patients <35 and 35–49 years of age at transplant who were either African-American or Hispanic (10/26 and 10/30 of these patients <35 and 35–49 years of age had GFNA), with estimated cumulative percentages developing GFNA (±SE) by 18 years post-transplant being 41.5% ± 10.2% and 36.9% ± 9.6%, respectively. For the other four groups defined by recipient age and race/ethnicity (2/94 of these patients combined had GFNA), the estimated cumulative percentages of developing GFNA by 18 years post-transplant ranged between 0.0 and 6.7%.

### 3.4. Characterization of NA Occurrence

During 18 years of post-transplant follow-up, 18.7% (28/150) of the cohort had become NA. As described above, 22/28 of these patients subsequently developed GFNA. A cumulative hazard plot of NA occurrence for all patients combined appears in Figure 4. Of note, a large one-year spike in NA occurrence occurred during the fourth year post-transplant, where 6/28 determinations of NA occurrence were made (apparently, right after 36 months post-transplant, when Medicare insurance coverage ended for all nondisabled transplant recipients <65 years of age). Interestingly, the hazard rate of NA occurrence (slope of the cumulative hazard curve) appeared to be reasonably constant beyond 5 years post-transplant (with no evidence of any falloff). Among the 28 patients who became NA, the median time to becoming NA was 46.2 months (range: 3.3–179.6 months) post-transplant; most of these determinations, 67.9% (19/28), occurred beyond 36 months post-transplant, respectively. Among the 22 patients who subsequently experienced GFNA, the median time from becoming NA until GFNA occurrence was 20.7 months (range: 2.4–118.6 months).

Cox stepwise regression of the hazard rate of becoming NA yielded the same two unfavorable factors as those found for the GFNA hazard rate: younger recipient age (as a continuous variable) (*p* = 0.000004) and African-American or Hispanic Recipient (*p* = 0.00007) (Cox model coefficients and corresponding SE’s are shown in Table 5). For this Cox model, no other baseline variable was significant in either univariable or multivariable analysis (results not shown). The observed percentages of patients who became NA, stratified by recipient age and race/ethnicity, were as follows: 42.9% (24/56) for African-American and Hispanic recipients <50 years of age, 7.7% (2/26) for non-African-American/non-Hispanic recipients <50 years of age, 6.5% (2/31) for African-American and Hispanic recipients ≥50 years of age, and 0.0% (0/37) for non-African-American/non-Hispanic recipients ≥50 years of age.

Descriptive characteristics for each of the 28 patients who became NA are shown in Table 6; 89.3% (25/28) developed a biopsy-proven acute rejection (BPAR) and/or chronic rejection (CR) episode (with T-cell, B-cell, or both types being present) as a direct consequence of their NA behavior, which also subsequently triggered GFNA in 22 patients. Time-to-BPAR (or CR) along with the corresponding histologic grade are shown for each of these 25 patients in Table 6; median time-to-BPAR/CR following the start of NA was 8.1 months (range: 0.6–118.2 months). Of note, the rejection episode directly due to NA behavior was the patient’s first BPAR in 15/25 cases, the patient’s second BPAR in 3/25 cases, and CR (without documented BPAR) in 7/25 cases. Histologic evidence of B-cell BPAR, i.e., acute antibody mediated rejection, was also present in 7/18 of the BPAR cases.

There were three patients who experienced rejection as a direct consequence of their NA behavior but without subsequently developing GFNA. Patient no. 10 had increasing renal dysfunction (i.e., serum creatinine) following her second BPAR (at 41.6 months), and then experienced DWFG due to a cardiovascular event at 55 months post-transplant. Patient no. 11, who admitted to having completely stopped taking her immunosuppressive medications twice, once for 4 weeks and then again for 8 weeks prior to developing a first BPAR at 103 months post-transplant, experienced a complete recovery of her renal function following treatment of the rejection episode. This patient became (and remained) reasonably compliant in taking her immunosuppressive medications thereafter and was alive with a completely functioning graft at 192 months post-transplant. Patient no. 23, following his first BPAR at 169.2 months, was alive with increasing renal dysfunction at 215 months post-transplant.

There were also three patients who became NA but without ever developing rejection as a direct consequence of their NA behavior. Patient no. 16, who became NA at 52.6 months post-transplant and exhibited NA behavior for approximately 16 months thereafter, never developed rejection and, subsequently, became (and remained) reasonably compliant in taking his immunosuppressive medications. This patient ultimately developed Other GF due to acute and chronic B-cell (antibody mediated) rejection at 164 months post-transplant. Patients no. 24 and no. 28 became NA at 120 months and 180 months post-transplant, respectively. While both patients remained intermittently NA thereafter, neither patient ever developed rejection nor suffered from any long-term renal dysfunction. These two patients were alive with completely functioning grafts at 204 months and 221 months post-transplant, respectively.

Among the 22 patients who developed GFNA, 21/22 continuously remained NA from the time of becoming NA until GFNA (see Table 6). Patient no. 4 was briefly NA for about 2 months starting at 15.3 months post-transplant (she experienced an empirically treated AR at 16.0 months post-transplant). While this patient was subsequently compliant in taking her immunosuppressive medications for approximately 3 years, she once again became (and remained) NA for about 26 months until experiencing a first BPAR followed by GFNA at 76.6 months and 78.3 months post-transplant, respectively. 

For each of the 28 NA cases, the attending physician determined that the patient had become NA in taking his/her immunosuppressive medications. Table 6 shows that the great majority of NA patients, 89.3% (25/28), had admitted to their NA behavior. In fact, 42.9% (12/28) of the NA patients admitted to having completely stopped taking their immunosuppressive medications; median length of time in which the patient admitted to having completely stopped taking his/her immunosuppression was 4 weeks (range: 1 week–18 months). In four additional cases, it was suspected that the patient had completely stopped taking his/her immunosuppressive medications prior to presenting with the BPAR (or CR). Overall, 11/28 NA patients displayed undetectable trough levels for at least one of his/her immunosuppressive medications; low and/or undetectable immunosuppression drug trough levels were documented in 26/28 NA patients.

Lastly, indicated and/or acknowledged reasons for the patient’s NA behavior included a strictly psychological component in 7/28 cases, and a combination of both financial and psychological components in most, 71.4% (20/28), of the cases. In only one case was the reason for the patient’s NA behavior not documented. Financial reasons included the patient’s expressed inability to pay for the immunosuppressive medications, including co-payments. Among the 20 patients having a documented financial component to their NA behavior, insurance issues were a contributing financial reason in 17/20 cases (4 patients were unemployed). Among the 27 patients with a documented psychological component to their NA behavior, clinical depression was previously diagnosed in 7/27 cases, and some combination of patient apathy, anxiety, and irritability (characteristics of clinical depression) was indicated in the remaining 20/27 cases.

## 4. Discussion

The results for this prospectively followed cohort of 150 adult, primary kidney transplant recipients at 18 years post-transplant validate our previous findings [1] in that: (i) GFNA was a major cause of graft loss, particularly among the higher risk subgroup identified as African-American and Hispanic recipients <50 years of age at transplant, with an estimated percentage of such patients ever developing GFNA ranging between 36.9 and 41.5%, (ii) This higher risk subgroup was also at higher risk of developing Other GF; thus, even if GFNA were to become eliminated as a cause of graft loss, African-American and Hispanic recipients <50 years of age would most likely still be identified as a higher risk subgroup for developing death-censored graft failure, and (iii) the major cause of graft loss among recipients ≥50 years of age at transplant was clearly DWFG; in fact, NA and GFNA rarely occurred among the older recipients.

In using our criteria for establishing NA behavior, 18.7% (28/150) had become NA, and 67.9% (19/28) of these determinations occurred beyond 36 months post-transplant. Similar to the results for GFNA, the 2 multivariable predictors of the hazard rate of NA occurrence were younger recipient age and African-American or Hispanic race/ethnicity. These results therefore confirm those previously reported by others regarding the associations of younger recipient age [2,3,7,13,16,20,21,38,39,40,41,42,43,44,45,46] and African-American/Hispanic race/ethnicity [2,3,9,38,39,42,47,48,49,50,51] with greater risk of patient NA. In addition, the lateness in observing overt NA behavior as reported by Dunn et al, [9] and the observation by Sellares et al. [10] that the majority of patients exhibiting overt NA behavior subsequently developed graft failure, were both confirmed by our results reported here.

As stated in the Introduction, many studies [13,14,15,16,17,18,19,20,21,22,23,24,25,26,27,28,29,30] have analyzed patient self-reporting results by focusing on “whether any immunosuppression-taking NA has occurred” (with “no” implying perfect adherence). In fact, the ordinal scale portion of the self-reporting questionnaire (BAASIS, ITAS, Morisky, or some other), which attempts to measure the extent of a patient’s NA behavior, has often not been more fully utilized. Furthermore, in some of these NA studies, the self-reporting questionnaire was only given to patients during the first 12–24 months post-transplant, and this study along with previous reports [1,2,3,4,5,6,7,8,9,10] have shown that both NA and GFNA are more likely to occur late, i.e., beyond 24 months, in the patient’s post-transplant follow-up. While use of a patient self-reporting questionnaire, as highlighted in these studies, [13,14,15,16,17,18,19,20,21,22,23,24,25,26,27,28,29,30] provides a simple snapshot of patient NA behavior, usually within the most recent 4 weeks (range: 1 week–3 months), achieving an accurate picture of each patient’s NA behavior over time would most likely require serial use of patient self-reporting throughout post-transplant follow-up.

We showed that in 89.3% (25/28) of the NA patients, a BPAR or CR occurred that was directly attributed to the patient’s NA behavior. In 21/22 of the GFNA cases, the NA behavior lasted continuously from the time of NA occurrence until GFNA. Thus, while most of our NA patients exhibited continuous NA behavior over time, and while NA occurrence was highly associated with subsequent risk of GFNA, our methodology used in determining the NA behavior was clearly imperfect, as 4/28 NA patients appeared to suffer no long-term consequences of their NA behavior (2/4 patients became and remained reasonably compliant after first exhibiting NA behavior). Again, these results highlight the importance of performing serial monitoring of patients (including the use of patient self-reporting) in determining the exact timing and extent of overt NA behavior over time. Our results clearly show that (i) such serial monitoring would need to be performed throughout patients’ complete post-transplant follow-up, and (ii) the determination of timing and extent of overt NA behavior is more important than simply focusing on whether or not any deviation from perfect patient adherence has occurred (e.g., 42.9% (12/28) of the NA patients in our study admitted to having completely stopped taking their immunosuppressive medications for a median length of 4 weeks).

As reported by Dunn et al, [9] many patients who become overtly NA in taking their immunosuppressive medications will also not follow laboratory test protocols. They will often not keep clinic appointments and may even cease communication with the transplant center for an extended period. Even if the patient admits to his/her NA behavior, as was the case for 89.3% (25/28) of our NA patients, it still becomes a difficult task for the transplant center to accurately document the patient’s timing and extent of overt NA behavior.

Finally, while a financial component to the patient’s NA behavior was documented in 71.4% (20/28) of the NA cases, an important psychological component to the patient’s NA behavior was also documented in 96.4% (27/28) of these cases. A similar finding among adult cases was also reported by Dunn et al. [9]; thus, a simple alleviation of the financial cost of immunosuppressive medications to kidney transplant recipients may not, by itself, completely overcome/avoid patient NA behavior. A few studies have shown that as Medicare’s immunosuppression coverage gets extended, income-related disparities in death-censored graft failure rates will diminish [52,53,54]. In fact, beginning in January 2023, all kidney transplant patients in the U.S. will received lifetime Medicare coverage of immunosuppression, regardless of when they received their kidney transplant [55]. While this Congressional mandate will certainly go a long way in helping all patients to be able to afford and better adhere to taking their immunosuppressive medications as prescribed, it is our belief (along with others [23,24,56,57,58]) that the psychological component to a patient’s NA behavior must also be simultaneously addressed in order to achieve long-lasting reductions in NA behavior, particularly in the higher risk subgroups. The use of a multicomponent intervention that includes a personal systems approach (i.e., a practical application of cognitive behavioral therapy) appears to offer promise [23,24,56,57,58] in helping organ transplant recipients better manage their behavior (psychological component) and, thus, overcome/avoid NA. Further follow-up of patients in these randomized trials [23,24,56,57,58] along with additional clinical studies are currently needed in determining whether these interventional approaches can be optimally used over a relatively short period (say, for 6–12 months) post-transplant vs. requiring its continued use throughout time post-transplant with the goal of achieving long-term “NA avoidance” efficacy.

## 5. Conclusions

These results highlight the importance of performing serial monitoring of patients for overt NA behavior throughout their post-transplant follow-up. Financial and psychological components to NA behavior need to be simultaneously addressed with the goal of achieving complete avoidance/elimination of NA behavior among higher risk patients.

## Figures and Tables

**Figure 1 jcm-11-01334-f001:**
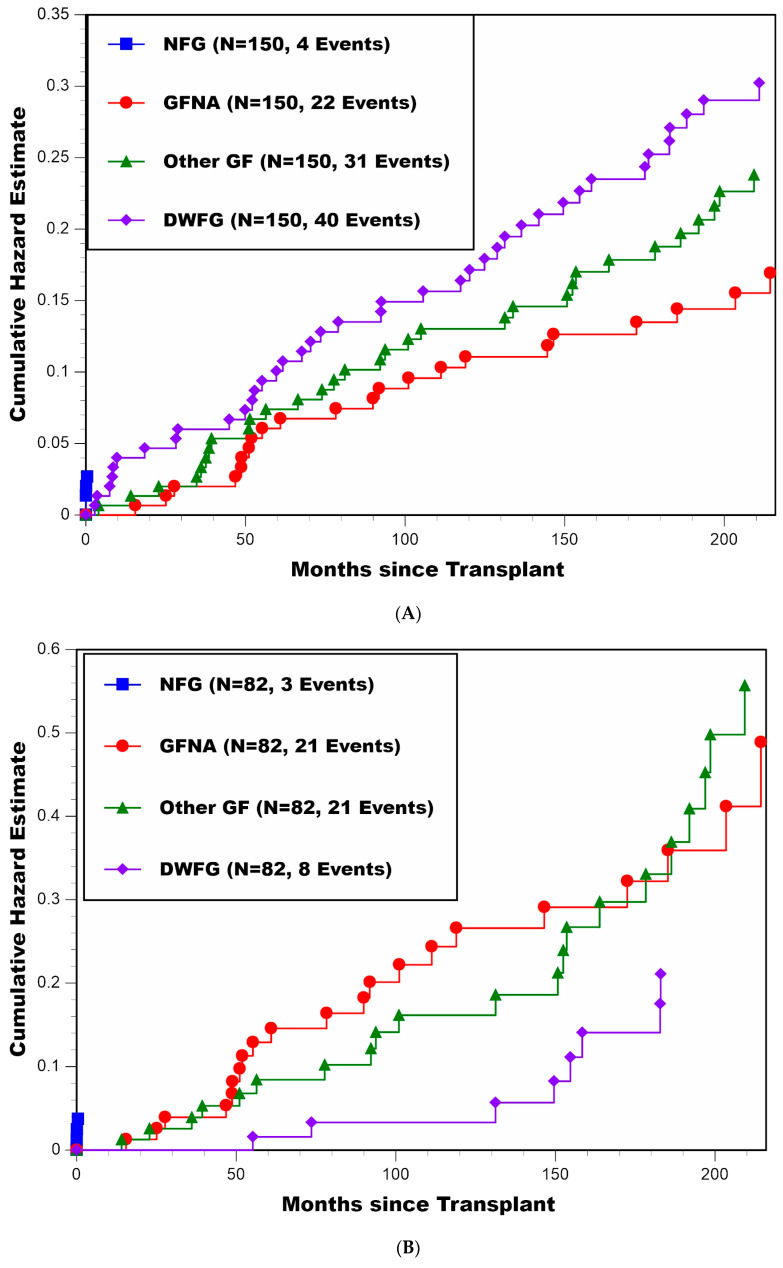
(**A**) Cumulative hazard plot of four distinct causes of graft loss (NFG, GFNA, Other GF, and DWFG) for all patients combined; (**B**) cumulative hazard plot of four distinct causes of graft loss (NFG, GFNA, Other GF, and DWFG) for patients <50 years of age at transplant; (**C**) cumulative hazard plot of four distinct causes of graft loss (NFG, GFNA, Other GF, and DWFG) for patients ≥50 years of age at transplant.

**Figure 2 jcm-11-01334-f002:**
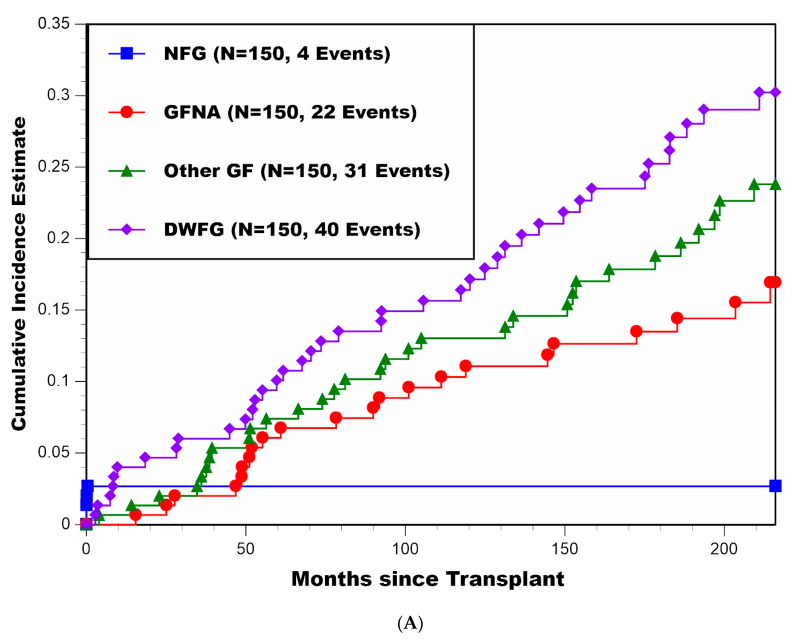
(**A**) Cumulative incidence estimates of four distinct causes of graft loss (NFG, GFNA, Other GF, and DWFG) for all patients combined; (**B**) cumulative incidence estimates of four distinct causes of graft loss (NFG, GFNA, Other GF, and DWFG) for patients <50 years of age at transplant; (**C**) cumulative incidence estimates of four distinct causes of graft loss (NFG, GFNA, Other GF, and DWFG) for patients ≥50 years of age at transplant.

**Figure 3 jcm-11-01334-f003:**
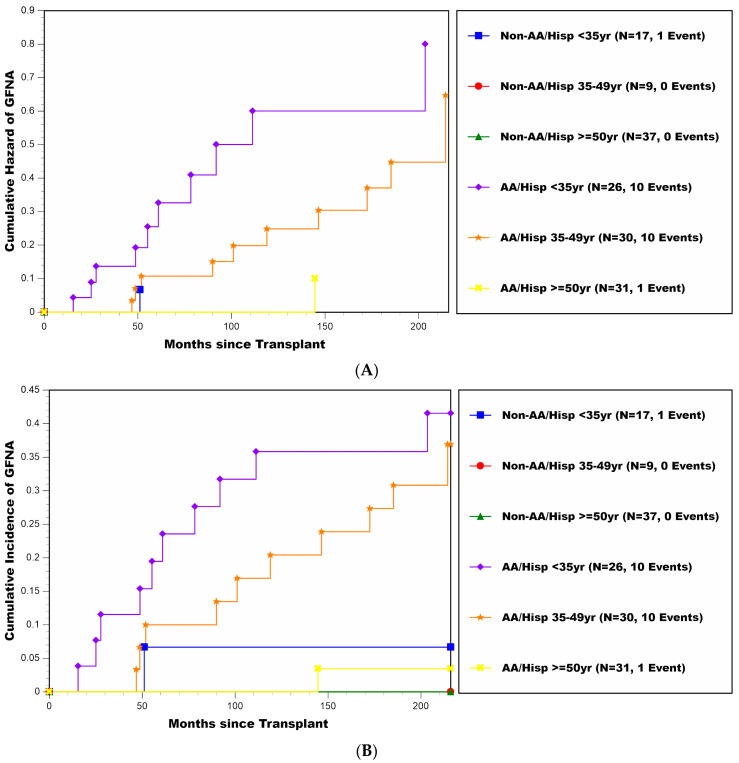
(**A**) Cumulative hazard plot of GFNA for patients stratified by recipient age at transplant (<35, 35–49, and ≥50 years) and race/ethnicity (African-American or Hispanic vs. non-African-American and non-Hispanic). (**B**) Cumulative incidence estimates of GFNA for patients stratified by recipient age at transplant (<35, 35–49, and ≥50 years) and race/ethnicity (African-American or Hispanic vs. non-African-American and non-Hispanic).

**Figure 4 jcm-11-01334-f004:**
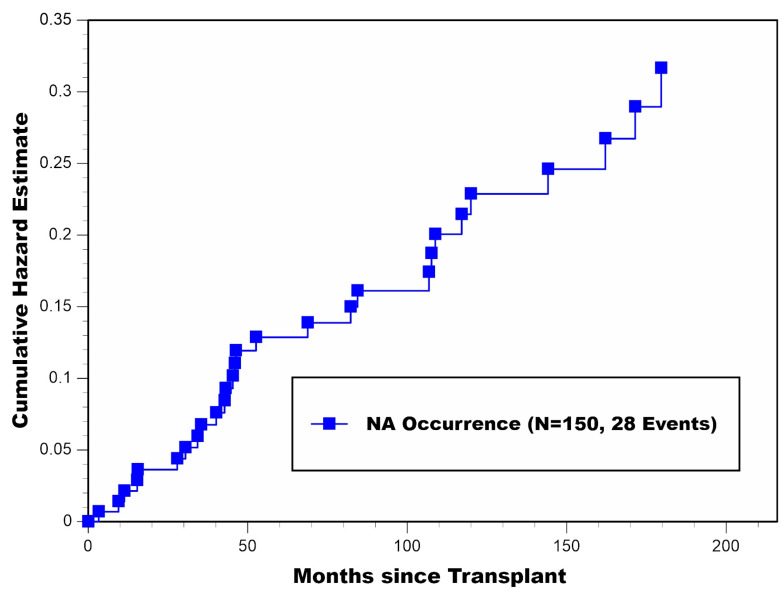
Cumulative hazard plot of NA occurrence for all patients combined.

**Table 1 jcm-11-01334-t001:** Distributions of selected baseline variables (N = 150).

Baseline Variable	Percentage with Characteristic if Categorical	Mean ± SE if Continuous
Recipient age (years)		46.9 ± 1.2 (N = 150)
		(Median = 48.0, range: 14–78)
Recipient gender		
Female	33.3% (50/150)	
Male	66.7% (100/150)	
Recipient race/ethnicity		
Black (non-Hispanic)	20.7% (31/150)	
Hispanic	37.3% (56/150)	
White (non-Hispanic) ^a^	42.0% (63/150)	
Recipient BMI (kg/m^2^)		25.6 ± 0.4 (N = 150)
		(Median = 25.0, range: 16.0–44.0)
Recipient pretransplant diabetes mellitus		
No	82.7% (124/150)	
Yes	17.3% (26/150)	
Recipient pretransplant history of cardiovascular disease		
No	81.3% (122/150)	
Yes	18.7% (28/150)	
Preemptive transplant		
No	88.0% (132/150)	
Yes	12.0% (18/150)	
Pretransplant time on dialysis (months) ^b^		29.1 ± 2.5 (N = 150)
		(Median = 20.7, range: 0–173.0)
Donor age (years)		37.7 ± 1.2 (N = 150)
		(Median = 37.5, range: 11–66)
Donor type		
LD	19.3% (29/150)	
DD	80.7% (121/150)	
CIT (h) among DD recipients		32.2 ± 0.7 (N = 121)
		(Median = 32, range: 17–59)
DCD status		
No	96.0% (144/150)	
Yes	4.0% (6/150)	
ECD status		
No	90.0% (135/150)	
Yes	10.0% (15/150)	
PRA >5%		
No	95.3% (143/150)	
Yes	4.7% (7/150)	
Total no. of HLA Mismatches		3.6 ± 0.1 (N = 150)
		(Median = 4, Range: 0–6)

Abbreviations: CIT, cold ischemia time; DCD, donation after cardiac death; DD, deceased donor; ECD, expanded criteria donor; HLA, human leukocyte antigen; LD, living donor; PRA, panel reactive antibodies. ^a^ Included among the White (non-Hispanic) subgroup were Southern Asian (N = 4), Middle Eastern (N = 2), and Native American Indian (N = 1) patients. ^b^ A value of 0.0 was used for preemptive transplant recipients.

**Table 2 jcm-11-01334-t002:** Cumulative incidence function (CIF) estimates of the percentages experiencing cause-specific graft loss.

**(a) All Patients Combined (N = 150)**				
Months	CIF: NFG	CIF: GFNA	CIF: Other GF	CIF: DWFG
Post-Transplant	% ± SE	% ± SE	% ± SE	% ± SE
60	2.7 ± 1.3	6.1 ± 2.0	7.4 ± 2.1	10.1 ± 2.5
120	2.7 ± 1.3	11.1 ± 2.6	13.0 ± 2.8	16.4 ± 3.1
180	2.7 ± 1.3	13.5 ± 2.9	18.8 ± 3.3	25.2 ± 3.7
216	2.7 ± 1.3	16.9 ± 3.4	23.8 ± 3.8	30.2 ± 4.1
**(b) Patients <50 Years of Age at Transplant (N = 82)**				
Months	CIF: NFG	CIF: GFNA	CIF: Other GF	CIF: DWFG
Post-Transplant	% ± SE	% ± SE	% ± SE	% ± SE
60	3.7 ± 2.1	11.2 ± 3.5	7.4 ± 2.9	1.3 ± 1.3
120	3.7 ± 2.1	20.5 ± 4.6	12.7 ± 3.8	2.6 ± 1.8
180	3.7 ± 2.1	23.4 ± 4.8	21.4 ± 4.8	8.3 ± 3.3
216	3.7 ± 2.1	28.8 ± 5.4	29.3 ± 5.4	11.3 ± 3.8
**(c) Patients ≥50 Years of Age at Transplant (N = 68)**				
Months	CIF: NFG	CIF: GFNA	CIF: Other GF	CIF: DWFG
Post-Transplant	% ± SE	% ± SE	% ± SE	% ± SE
60	1.5 ± 1.5	0.0 ± 0.0	7.4 ± 3.2	20.6 ± 4.9
120	1.5 ± 1.5	0.0 ± 0.0	13.4 ± 4.1	32.7 ± 5.7
180	1.5 ± 1.5	1.7 ± 1.7	15.1 ± 4.4	45.7 ± 6.4
216	1.5 ± 1.5	1.7 ± 1.7	15.1 ± 4.4	55.1 ± 7.2

**Table 3 jcm-11-01334-t003:** Cox model results for the: (i) hazard rate of developing GF due to nonadherence, and (ii) subdistribution hazard of GF due to nonadherence (22 events each).

**(i) Hazard Rate of GF due to Nonadherence**			
	Univariable	Multivariable	Multivariable Model ^2^
Baseline Variable ^1^	*p*-value	*p*-value	Coeff ± SE
Recipient age	0.00007	(√) 0.0001	−0.065 ± 0.018
Afr-Am or Hisp recipient	0.0004	(√) 0.0007	2.647 ± 1.025
**(ii) Subdistribution Hazard of GF due to Nonadherence**			
	Univariable	Multivariable	Multivariable Model ^2^
Baseline Variable ^1^	*p*-value	*p*-value	Coeff ± SE
Recipient age	0.000005	(√) 0.00003	−0.067 ± 0.017
Afr-Am or Hisp recipient	0.0003	(√) 0.001	2.575 ± 1.024

Abbreviations: Afr-Am, African-American; Coeff, coefficient; Hisp, Hispanic. ^1^ The two variables considered for each Cox model were defined as follows: recipient age (continuous variable); and Afr-Am or Hisp recipient = {1 if recipient race/ethnicity was African-American or Hispanic, 0 otherwise}. ^2^ For both Cox models, the order of selection was as follows: recipient age, then Afr-Am or Hisp recipient. For both Cox models, no other baseline variable was significant in either univariable or multivariable analysis (results not shown). (√) represents selection into the Cox model.

**Table 4 jcm-11-01334-t004:** Cumulative incidence function (CIF) estimates of the percentages experiencing GFNA, stratified by recipient age (<35, 35–49, and ≥50 years) and recipient race/ethnicity (non-African-American and non-Hispanic vs. African-American or Hispanic).

**(a) Non-African-American and Non-Hispanic Recipients (N = 63)**			
	Recipient Age		
Months	<35 years (N = 17)	35–49 years (N = 9)	≥50 years (N = 37)
Post-Transplant	% ± SE	% ± SE	% ± SE
60	6.7 ± 6.4	0.0 ± 0.0	0.0 ± 0.0
120	6.7 ± 6.4	0.0 ± 0.0	0.0 ± 0.0
180	6.7 ± 6.4	0.0 ± 0.0	0.0 ± 0.0
216	6.7 ± 6.4	0.0 ± 0.0	0.0 ± 0.0
**(b) African-American or Hispanic Recipients (N = 87)**			
	Recipient Age		
Months	<35 years (N = 26)	35–49 years (N = 30)	≥50 years (N = 31)
Post-Transplant	% ± SE	% ± SE	% ± SE
60	19.5 ± 7.8	10.0 ± 5.5	0.0 ± 0.0
120	35.8 ± 9.6	20.4 ± 7.4	0.0 ± 0.0
180	35.8 ± 9.6	27.4 ± 8.3	3.4 ± 3.4
216	41.5 ± 10.2	36.9 ± 9.6	3.4 ± 3.4

**Table 5 jcm-11-01334-t005:** Cox model results for the hazard rate of becoming nonadherent (28 events).

	Univariable	Multivariable	Multivariable Model ^2^
Baseline Variable ^1^	*p*-value	*p*-value	Coeff ± SE
Recipient age	0.000003	(√) 0.000004	−0.072 ± 0.017
Afr-Am or Hisp recipient	0.00006	(√) 0.00007	2.372 ± 0.738

Abbreviations: Afr-Am, African-American; Coeff, coefficient; Hisp, Hispanic. ^1^ The two variables selected into the Cox model were defined as follows: recipient age (continuous variable); and Afr-Am or Hisp recipient = {1 if recipient race/ethnicity was African-American or Hispanic, 0 otherwise}. ^2^ The order of selection into this Cox model was as follows: recipient age, then Afr-Am or Hisp recipient. For this Cox model, no other baseline variable was significant in either univariable or multivariable analysis (results not shown). Note: (√) Represents Selection into the Cox Model.

**Table 6 jcm-11-01334-t006:** Descriptive characteristics for each of the 28 patients who became nonadherent (NA) (listed by shortest to longest time to NA).

							Approximate		Documented	Indicated/
							Length	Patient	Low/	Acknowledged
				Month to First		Month from	of Time	Admits	Undetectable	Reason(s) for
	Age at	Race/	Month to	BPAR/CR	Month to	NA to	with NA	to the NA	Trough	the NA
Patient	Tx	Ethnicity	NA ^1^	(Grade) ^1^	GFNA	GFNA ^1^	Behavior	Behavior	Levels	Behavior ^9^
#1	34	Black	3.3	4.0 (IB) ^2^	15.5	12.2	12.2	Y	U	F/P
#2	32	Black	9.5	10.1 (IIA)	25.1	15.6	15.6	Y ^8^	L	P
#3	31	White	11.3	51.1 (CR) ^4^	51.1	39.8	39.8	Y	L	F(Ins)/P
#4	29	Black	15.3	76.6 (IB) ^3^	78.3	63.0	2.0 + 26.0	Y ^7^	L	F(Ins)/P
#5	31	Black	15.5	17.2 (IIA)	27.7	12.2	12.2	Y ^8^	L	F/P(CDep)
#6	36	Hispanic	27.9	146.1 (Bord) ^2^	146.5	118.6	118.6	Y ^8^	L & U	F/P(CDep)
#7	14	Black	30.5	32.1 (IIA) ^3^	48.8	18.3	18.3	N	L	P
#8	45	Hispanic	34.3	40.4 (IA)	46.8	12.5	12.5	Y ^7^	L	P
#9	33	Hispanic	35.4	41.3 (IB)	91.8	56.4	56.4	Y ^8^	L & U	F(Ins)/P
#10	46	Hispanic	40.1	41.6 (IB) ^3^	----- ^5^	----- ^5^	14.9	Y ^8^	L	F(Ins)/P
#11	15	Black	42.7	103.0 (IB) ^2^	----- ^5^	----- ^5^	1.0 + 2.0	Y^8^	L & U	F(Ins)/P(CDep)
#12	44	Hispanic	43.1	51.2 (IB)	51.9	8.8	8.8	Y ^8^	L	F(Ins)/P
#13	17	Hispanic	45.3	48.3 (IB)	60.9	15.6	15.6	Y ^8^	L & U	P(CDep)
#14	22	Hispanic	46.0	53.3 (IA) ^3^	55.2	9.2	9.2	Y ^8^	L & U	F(Ins)/P
#15	40	Hispanic	46.3	48.7 (CR) ^4^	48.7	2.4	2.4	Y ^7^	N/A	F(Ins)/P
#16	33	White	52.6	-----	----- ^6^	----- ^6^	16.0	Y	L & U	F(Ins)/P
#17	40	Hispanic	68.8	79.4 (Bord)	101.1	32.3	32.3	Y ^8^	L & U	F(Ins)/P(CDep)
#18	37	Hispanic	82.3	89.9 (CR) ^4^	89.9	7.6	7.6	Y ^8^	N/A	P
#19	19	Hispanic	84.4	111.2 (CR) ^3^	111.2	26.8	26.8	Y	L	F(Ins)/P
#20	53	Hispanic	106.8	141.8 (Bord) ^2^	144.6	37.8	37.8	Y	L	F(Ins)/P
#21	37	Hispanic	107.6	214.4 (CR) ^4^	214.4	106.8	106.8	Y	L&U	F(Ins)/P(CDep)
#22	41	Black	108.8	109.5 (IA) ^2^	118.9	10.1	10.1	Y ^8^	L	P
#23	25	Hispanic	117.1	169.2 (IIA) ^2^	----- ^5^	----- ^5^	97.9	Y ^7^	L	F(Ins)/P(CDep)
#24	45	Black	120.0	-----	----- ^6^	----- ^6^	84.0	Y	L	P
#25	38	Hispanic	144.1	172.5 (CR) ^4^	172.5	28.4	28.4	Y	L	F(Ins)/P
#26	39	Black	162.1	185.2 (CR) ^4^	185.2	23.1	23.1	N/A	U	N/A
#27	27	Hispanic	171.5	193.2 (Bord) ^2^	203.5	32.0	32.0	N/A	L	F(Ins)/P
#28	56	Hispanic	179.6	-----	----- ^6^	----- ^6^	41.4	Y	L & U	F(Ins)/P

Abbreviations: Bord, borderline; BPAR, biopsy-proven acute rejection; CDep, clinically depressed; CR, chronic rejection; N/A, not available; F, financial; Ins, insurance issues; P, psychological. ^1^ Median time-to-NA, among the 28 patients who became NA, was 46.2 (range: 3.3–179.6) months. Of note, it was determined that in 25/28 patients who became NA, they developed an acute and/or chronic rejection episode (with T-cell, B-cell, or both types being present) as a direct consequence of their NA behavior, which also subsequently triggered kidney allograft failure in all 22 patients who developed GFNA. Among the 22 patients who developed GFNA, the median time from NA occurrence to development of GFNA was 20.7 (range: 2.4–118.6) months post-transplant. ^2^ The patient also had histologic evidence of acute AMR at this time. ^3^ The BPAR shown here was, in fact, the second BPAR for patient nos. 7, 10, and 14. In addition, patient no. 4 developed an empirical AR (clinically indicated and treated, but no biopsy was performed) at 16 months, 3 weeks after being documented as having become NA. Lastly, the NA date for patient no. 19 occurred at approximately one month following a first BPAR. ^4^ Patient nos. 3, 15, 18, 21, 25, and 26, while never developing BPAR, had ongoing chronic (and probably acute) rejection due to NA at the time of returning to permanent dialysis (no biopsy was performed at that time). ^5^ There were three patients who experienced rejection as a direct consequence of their NA behavior but without subsequently developing GFNA. Patient no. 10 had increasing renal dysfunction (i.e., serum creatinine) following her second BPAR (at 41.6 months), and then experienced DWFG due to a cardiovascular event at 55 months post-transplant. Patient no. 11 was alive with a completely functioning graft at the time of being lost to follow-up at 192 months post-transplant. Patient no. 23, following his first BPAR at 169.2 months, was alive with increasing renal dysfunction (i.e., serum creatinine) at last follow-up of 215 months post-transplant. ^6^ There were three patients who became NA but without ever developing rejection as a direct consequence of their NA behavior. Patient no. 16, who became NA at 52.6 months post-transplant and exhibited NA behavior for approximately 16 months thereafter, never developed rejection and subsequently became (and remained) reasonably compliant in taking his immunosuppressive medications. This patient ultimately developed Other GF due to acute and chronic B-cell (antibody mediated) rejection at 164 months post-transplant. Patient no. 24, who became NA at 120 months post-transplant and remained intermittently NA thereafter, never developed rejection or suffered from any long-term renal dysfunction. He became lost to follow-up at 204 months post-transplant (with a completely functioning graft at last follow-up). Lastly, patient no. 28, who became NA at 180 months post-transplant and remained intermittently NA thereafter, also never developed rejection or suffered from any long-term renal dysfunction. He was alive with a completely functioning graft at last follow-up of 221 months post-transplant. ^7^ It was suspected that patient nos. 4, 8, 15, and 23 had completely stopped taking their immunosuppressive medications prior to presenting with the first BPAR/CR. ^8^ Twelve patients admitted/acknowledged that they had completely stopped taking their immunosuppressive medications prior to presenting with the first BPAR (second BPAR for patient nos. 10 and 14; CR and immediate graft failure for patient no. 18). The approximate lengths of time that each of these patients had completely stopped taking their immunosuppressive medications prior to presenting with the first BPAR (again, second BPAR for patient nos. 10 and 14, and CR/immediate graft failure for patient no. 18) were as follows: patient no. 2: 5 days; patient no. 5: 6 weeks; patient no. 6: 18 months; patient no. 9: 2–3 weeks; patient no. 10: 3 weeks; patient no. 11: 4 weeks and 8 weeks; patient no. 12: 3 months; patient no. 13: 2–3 weeks; patient no. 14: 3 months; patient no. 17: 4 weeks; patient no. 18: 6 months; and patient no. 22: 3 weeks. Including the two distinct episodes for patient no. 11 as separate events, the median time that these patients admitted to having completely stopped taking their immunosuppressive medications was 4 weeks (range: 5 days–18 months). ^9^ Among the 27 patients who had documented (indicated or acknowledged) psychological components to their NA behavior, clinical depression was previously diagnosed in 7/27 cases, and a combination of patient apathy, anxiety, and irritability (characteristics of clinical depression) was indicated in the remaining 20/27 cases.

## Data Availability

De-identified data will be made available upon request.

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
