# Peer review of "Graft Failure Due to Nonadherence among 150 Prospectively-Followed Kidney Transplant Recipients at 18 Years Post-transplant: Our Results and Review of the Literature"

_jcm, 2022, doi:10.3390/jcm11051334_

Round 1

Reviewer 1 Report

Dear Dr Gaynor

your paper is written in a very understandable manner. The long follow-up as well as the representative number of patients  are plus points. Based on the analysis GFNA, particulary among African/Americans and Hispanics < 50 years was a major cause of graft loss besides „other GF“. In elderly patients (>50 years) DWFG was the main reason for graft loss.

Some minor questions remain:

 -as the authors stated examined pts. have been included in a trial with 3 different regimens of immunosuppression (tacrolimus/sirolimus, tacrolimus/MPA, cyclosporine/sirolimus): did underlying immunosuppressive regimens had any impact on detected results? Especially was „other GF“ triggered by underlying immunosuppression, because e.g. the combination of tacrolimus/MPA might be in favor for some renal diseases, e.g. lupus nephritis, which might be more frequent in  „younger African-Americans“

- if used immunosuppression had any impact on detected results, immunosuppression has to be included in the uni-/multivariable models

- as quote 11, being published in Clinical Transplantation 2020 on which the authors refer to is not an open access paper, at least some of the baseline characteristics of the pts. population should be included in current publication

Author Response

Point by point responses are attached. Thank you.

Reviewer 2 Report

This is an interesting analysis of graft loss to nonadherence many years after kidney transplantation.  It is well done and represents a novel contribution to the literature.

Author Response

Thank you very much.